# Effects of Individual Circulating FFAs on Plasma and Hepatic FFA Epoxides, Diols, and Epoxide-Diol Ratios as Indices of Soluble Epoxide Hydrolase Activity

**DOI:** 10.3390/ijms241310760

**Published:** 2023-06-28

**Authors:** Young Taek Oh, Jun Yang, Christophe Morisseau, Qiyi He, Bruce Hammock, Jang H. Youn

**Affiliations:** 1Department of Physiology and Neuroscience, University of Southern California Keck School of Medicine, 2250 Alcazar Street, CSC 214, Los Angeles, CA 90089, USA; youngoh@usc.edu; 2Department of Entomology and Nematology, University of California, Davis, CA 95616, USA; junyang@ucdavis.edu (J.Y.); chmorisseau@ucdavis.edu (C.M.); qiyhe@ucdavis.edu (Q.H.); bdhammock@ucdavis.edu (B.H.)

**Keywords:** lipid infusion, olive oil, safflower seed oil, fish oil, ω-3 fatty acids

## Abstract

Oxylipins, oxidation products of unsaturated free fatty acids (FFAs), are involved in various cellular signaling systems. Among these oxylipins, FFA epoxides are associated with beneficial effects in metabolic and cardiovascular health. FFA epoxides are metabolized to diols, which are usually biologically less active, by soluble epoxide hydrolase (sEH). Plasma epoxide-diol ratios have been used as indirect measures of sEH activity. This study was designed to examine the effects of acute elevation of individual plasma FFAs on a variety of oxylipins, particularly epoxides, diols, and their ratios. We tested if FFA epoxide-diol ratios are altered by circulating FFA levels (i.e., substrate availability) independent of sEH activity. Wistar rats received a constant intravenous infusion of olive (70% oleic acid (OA)), safflower seed (72% linoleic acid (LA)), and fish oils (rich in ω-3 FFAs) as emulsions to selectively raise OA, LA, and ω-3 FFAs (eicosapentaenoic acid (EPA) and docosahexaenoic acid (DHA)), respectively. As expected, olive, safflower seed, and fish oil infusions selectively raised plasma OA (57%), LA (87%), EPA (70%), and DHA (54%), respectively (*p* < 0.05 for all). Raising plasma FFAs exerted substrate effects to increase hepatic and plasma epoxide and diol levels. These increases in epoxides and diols occurred to similar extents, resulting in no significant changes in epoxide-diol ratios. These data suggest that epoxide-diol ratios, often used as indices of sEH activity, are not affected by substrate availability or altered plasma FFA levels and that epoxide-diol ratios may be used to compare sEH activity between conditions of different circulating FFA levels.

## 1. Introduction

Unsaturated free fatty acids (FFAs) are oxidized to produce a variety of oxylipins, many of which are involved in various cellular processes, including pro- and anti-inflammatory responses [1,2,3]. The biosynthesis of oxylipins involves three families of enzymes: cytochrome P450 (CYP), cyclooxygenase (COX), and lipoxygenase (LOX). FFA epoxides, produced by CYP, are known to be associated with beneficial effects in metabolic and cardiovascular health [4,5,6]. FFA epoxides are metabolized to usually biologically less active diols largely by soluble epoxide hydrolase (sEH). Inhibition of sEH, which increases FFA epoxides, improves glucose homeostasis and cardiovascular health [4,5,6] and is proposed as an effective strategy to treat diabetes and cardiovascular diseases [1,7]. Recent human studies have demonstrated significant associations of sEH polymorphism with various metabolic or cardiovascular diseases [8,9,10,11,12], suggesting the important role of sEH in human physiology and diseases. Recent studies have expanded the potential clinical paths of sEH inhibition to lung, kidney, and neural diseases [13,14,15].

Despite its importance to metabolic and cardiovascular health, its regulation in vivo is poorly understood, particularly in humans, where sEH is difficult to study, largely due to limited access to individual tissues where sEH is expressed. sEH activity is often inferred in human studies by measuring FFA epoxide-diol ratios in the blood [16,17]. Epoxides and diols derive from precursor (or substrate) FFAs, and, therefore, epoxide and diol levels may be altered by the availability of substrate FFAs, but it is unknown whether epoxide-diol ratios are affected by FFA availability. This is an important issue, especially when epoxide-diol ratios are compared as indices of sEH activity between conditions with different circulating FFA levels. For example, when these ratios are altered in obesity or high-fat-fed animals, the possibility may exist that some of the observed effects arose simply from different circulating FFA levels, independent of changes in sEH activity. This critical issue has not been directly addressed, despite the increasing use of plasma FFA epoxide-diol ratios (or epoxide and diol levels) as indices of sEH activity in human studies [16,17,18,19,20].

In the present study, we will address this issue by examining the effects of acute elevation of individual circulating FFAs on epoxide and diol levels and their ratios in plasma and the liver. To achieve this goal, we selectively raised circulating linoleic (LA) and oleic (OA) acids by infusing safflower seed (72% LA) and olive (70% OA) oils, respectively, as in our previous study [21,22]. LA is one of the precursor FFAs for FFA epoxides and diols, and, therefore, we should be able to evaluate its mass-action effects to alter its epoxides, diols, or their ratios. In contrast, OA was not metabolized to epoxides and diols detectable in our oxylipin assays, and the experiments with olive oil infusion will serve as a control to examine non-mass-action effects, if any, of circulating FFA to alter epoxides and diols derived from other precursor FFAs. Furthermore, fish oil, rich in ω-3 FFAs, was infused to elevate plasma levels of eicosapentaenoic (EPA) and docosahexaenoic (DHA) acids. After the oil infusions, plasma and liver samples were collected and analyzed for oxylipins, and, using these data, we tested the hypothesis that altered precursor FFA levels may alter their epoxide and diol levels but not epoxide-diol ratios, which may reflect sEH activity independent of circulating (or intracellular) FFA levels.

## 2. Results

### 2.1. Effects of Oil Infusions on Individual Circulating FFA Levels

Olive, safflower seed, and fish oils were infused intravenously as emulsion into rats for 2 h, and blood samples were collected before and after the infusions and analyzed for total and individual FFAs. As expected, olive oil, predominantly composed of OA (C18:1), selectively raised plasma OA by 57% (*p* < 0.001 vs. control; Figure 1A) without altering other FFAs. Similarly, safflower seed oil, predominantly composed of LA (C18:2), selectively raised plasma LA by 87% (*p* < 0.001; Figure 1B) without altering other FFAs. Furthermore, fish oil, rich in ω-3 FFAs, raised plasma EPA (C20:5; 70%, *p* = 0.02) and DHA (C22:6; 54%, *p* < 0.001), together with arachidonic acid (ARA; C20:4; 31%, *p* < 0.001) levels (Figure 1C,E). In contrast, plasma palmitate (C16:0) and α-linolenic acid (ALA; C18:3) levels were not altered by the lipid infusions (Figure 1F,G). The total FFA levels, measured by an enzymatic assay, were increased by 16% (*p* < 0.05), 42% (*p* < 0.001), and 21% (*p* < 0.001) by the olive, safflower seed, and fish oil infusions, respectively (Figure 1H). Thus, the olive and safflower seed oil infusions selectively raised plasma OA and LA levels, respectively, as in our earlier study [21,22], and fish oil raised the ω-3 FFAs EPA and DHA, together with ARA.

### 2.2. Effects of Oil Infusions on Hepatic Epoxide and Diol Levels and Epoxide-Diol Ratios

We next examined the effects of the oil infusions on oxylipins, particularly on FFA epoxide and diol levels, in the liver, the major tissue for lipid metabolism. To obtain control liver samples, the oil infusion experiments were expanded to have four experimental groups: the control and the three oil infusion groups. In addition, the infusion period was extended to 3 h to allow enough time for the manifestation of the oil infusion effects. At the end of each infusion, plasma and liver samples were collected for oxylipin analysis. Figure 2 shows changes in hepatic epoxides, diols, and their ratios, derived from different precursor FFAs, with the oil infusions. First of all, hepatic levels of epoxides and diols derived from ALA were not altered by either of the oil infusions (*p* > 0.05; Figure 2A,B), consistent with unaltered plasma ALA levels with the oil infusions. Olive oil infusion, which increased only plasma OA, did not significantly alter epoxides or diols derived from ALA, LA, ARA, EPA, or DHA (i.e., 13 epoxide-diol pairs), or their ratios, compared to the control. In contrast, safflower seed oil infusion, which selectively increased plasma LA by 87%, increased two epoxides derived from LA (12(13)-EpOME and 9(10)-EpOME) by ~2.5-fold (Figure 2D), indicating mass-action effects. These increases in LA epoxides were accompanied by ~2.1-fold and ~5.7-fold increases in corresponding diols (12,13-DiHOME and 9,10-DiHOME; Figure 2E), resulting in no change or a 2.1-fold increase (*p* < 0.05) in the diol-to-epoxide ratio (Figure 2F). Fish oil infusion, which increased plasma DHA and EPA levels by 54% and 70%, respectively, increased DHA epoxides (4 species; see Figure 2 legend) by ~2.5- to ~4-fold (*p* < 0.001; Figure 2J) and EPA epoxides (3 species) by 2.5- to 8.9-fold (*p* < 0.01; Figure 2M), again indicating mass-action effects. These increases in DHA and EPA epoxides were accompanied by similar increases in DHA and EPA diols (*p* < 0.001; Figure 2K,N), resulting in no significant changes in diol-to-epoxide ratios for the DHA and EPA epoxide-diol pairs (Figure 2L,O). Thus, fish oil infusion affected DHA and EPA epoxides and diols to similar extents, and epoxide-diol ratios were not significantly altered from control for all 7 ω-3 FFA epoxide-diol pairs.

### 2.3. Effects of Oil Infusions on Plasma FFA Epoxide and Diol Levels and Epoxide-Diol Ratios

In the present study, plasma oxylipin assays were more variable than those for hepatic oxylipins, and we increased the number of samples to 12 and discussed only epoxide-diol pairs reliably detected in most (>90%) samples. Although fewer epoxide-diol pairs were analyzed (Figure 3), the patterns were similar to those with hepatic oxylipins. First, hepatic levels of an ALA epoxide (15(16)-EpODE) and its corresponding diol (15,16-DiHODE) were not altered by either of the oil infusions (*p* > 0.05; Figure 3A,B), consistent with unaltered plasma ALA levels. Olive oil infusion did not significantly alter epoxide or diol levels derived from ALA, LA, or DHA, consistent with hepatic epoxide/diol levels with the oil infusion. In contrast, safflower seed oil infusion, which selectively increased plasma LA by 87%, increased two epoxides derived from LA (12(13)-EpOME and 9(10)-EpOME) by ~3- and ~7-fold (*p* < 0.01; Figure 3D), respectively, indicating mass-action effects. These increases in LA epoxides were accompanied by similar increases in corresponding diols (12,13-DiHOME and 9,10-DiHOME; *p* < 0.01; Figure 3E) and resulted in no changes in the epoxide-diol ratio (Figure 3F). Fish oil infusion, which increased plasma DHA levels by 54%, increased two DHA epoxides (16(17)-EpDPE and 19(20)-EpDPE) by ~6- and ~7-fold (*p* < 001; Figure 3G), respectively, again indicating mass-action effects. These increases in DHA epoxides were accompanied by ~2-fold increases in DHA diols, although statistically insignificant, resulting in no significant changes in epoxide-diol ratios for the DHA epoxide-diol pairs (Figure 3H,I). Thus, none of the five epoxide-diol pairs showed significant changes in epoxide-diol ratios with the oil infusions.

### 2.4. Effects of Oil Infusions on SAMI-Estimated and Directly Measured sEH Activity

We previously introduced a method for estimating sEH activity based on multiple epoxide-diol ratios, called SAMI (simultaneous assessment of multiple indices) [23]. Using this method, we estimated sEH activity based on all available plasma (*n* = 13) or liver (*n* = 5) epoxide-diol ratios. Figure 4 shows that SAMI-estimated sEH activity was not altered by the different oil infusions, regardless of whether it was assessed based on plasma or hepatic data, suggesting that raising plasma (total or individual) FFA levels did not affect sEH activity, reflected on epoxide-diol ratios. We next examined the effects of the oil infusions on sEH activity and protein levels directly measured in the liver, as described in the Materials and Methods section. Figure 5 shows that neither sEH activity nor sEH protein level was altered in the liver by the different oil infusions, consistent with no significant effects of the oil infusions on plasma and hepatic epoxide-diol ratios or SAMI-estimated sEH activity.

### 2.5. Effects of Oil Infusions on Hepatic Oxylipins Produced by LOX and COX

We next examined if the oil infusions affected other oxylipins produced by LOX and COX. Safflower oil infusion, which increased plasma LA levels by ~1.9-fold, increased hepatic 9-HODE levels by ~23-fold (*p* < 0.001; Figure 6), which was larger than other oxylipins derived from LA. Fish oil infusion, which increased plasma EPA by ~1.7-fold, increased hepatic levels of HEPEs, produced from EPA by LOX by 5- to 21-fold (*p* < 0.001), suggesting that the availability of EPA may be limiting the production of these oxylipins. Hepatic levels of PGD3, produced from EPA by COX, also increased ~8-fold with fish oil infusion (*p* < 0.001), again suggesting that the production of this metabolite is limited by substrate availability. These are such important observations (see Section 3).

## 3. Discussion

The present study shows that intravenous lipid infusions, which selectively increased individual plasma FFAs, increased hepatic levels of FFA epoxides derived from these precursor FFAs, indicating mass-action (or substrate) effects of plasma FFA on the production of hepatic epoxides. In addition, increased hepatic FFA epoxides were associated with similar increases in corresponding diols, suggesting that the production of FFA diols by sEH also depended on the availability of their precursors, the FFA epoxides. The increases in hepatic levels of FFA epoxides and diols were similar in most epoxide-diol pairs, resulting in no significant changes in epoxide-diol ratios. Similar patterns were observed with plasma FFA epoxides and diols, and elevating plasma FFAs did not significantly alter plasma epoxide-diol ratios, suggesting that these ratios, often used as indirect measures of sEH activity [16,17], may not be altered simply by substrate availability or altered plasma FFA levels. This finding is important, assuring that epoxide-diol ratios can be compared as indices of sEH activity between conditions of different FFA levels.

Raising plasma levels of LA (via safflower oil infusion) raised plasma and hepatic levels of epoxides derived from LA but not those from other FFAs. Similarly, raising plasma levels of EPA and DHA (via fish oil infusion) raised plasma and hepatic levels of epoxides derived from EPA and DHA, respectively, but not those from other FFAs. Thus, the effects of FFAs to increase hepatic epoxide levels were specific to those derived from these precursor FFAs. In contrast, raising plasma levels of OA did not alter any FFA epoxides from LA, EPA, and DHA. These patterns, observed with all hepatic (*n* = 13) and plasma (*n* = 5) epoxides examined, clearly demonstrate that the increases in epoxides were due to mass-action (or substrate) effects, not to non-mass-action effects of FFAs on the production (or degradation) of epoxides, and the production of FFA epoxides by CYP depended on the availability of precursor FFAs. Interestingly, the increases in FFA epoxides were generally greater in fold effect than the increases in plasma precursor FFAs. CYP competes for FFAs with other enzymes (e.g., COX and LOX), and increased FFA availability would differentially impact fluxes through these enzymes depending on their enzyme kinetics. Our data suggest that the flux through CYP was more sensitive to substrate availability than other fluxes of FFA oxidation or metabolism. In general, the ω-3 fatty acids EPA and DHA are poor substrates, unlike arachidonate for inflammatory COX pathways.

Another common feature observed with most epoxide-diol pairs is that increases in FFA epoxides were associated with similar increases in corresponding diols, suggesting that fluxes through sEH to produce diols were also limited by the availability of their substrates (i.e., FFA epoxides), and when more substrates were available, the fluxes through sEH also increased. Interestingly, hepatic diol levels did not increase to a greater extent than those in epoxides (in all but one epoxide-diol pair; see below), suggesting diol elimination processes (e.g., via sulfation, glucuronidation, or simple glomerular filtration) also increased similarly. Thus, under these conditions of increased fluxes through sEH, intermediate levels were similarly increased in response to increased substrate availability (or showed similar sensitivities to substrate availability), not altering their ratios, which may be determined by the kinetic properties of enzymes involved. Thus, similar increases in FFA epoxides and diols resulted in unaltered epoxide-diol ratios, indicating that increased fluxes through the CYP–sEH pathway did not change the steady state balances among intermediates. Thus, the epoxide-diol ratios may be relatively insensitive to the flux through CYP (converting FFAs to epoxides) or sEH (converting epoxides to diols), at least under the present experimental conditions (i.e., 16–87% increases in total or individual plasma FFAs; Figure 1). These data suggest that epoxide-diol ratios may reflect sEH activity, independent of substrate availability (or FFA levels) or flux through the CYP–sEH pathway.

Not all epoxide pairs behaved following the general patterns described above. For example, raising plasma LA increased hepatic levels of 9(10)-EpOME, an epoxide derived from LA, and its corresponding diol 9,10-DiHOME, but the increase in diol (~5.7-fold) was greater than that in epoxide (~2.5-fold), resulting in a significant change in epoxide-diol ratio (Figure 2D–F). Also, raising plasma DHA increased plasma levels of epoxides and diols, but the increases were statistically significant for epoxides but not for diols (Figure 3G,H). Although these deviations from the general patterns may reflect factors associated with specific epoxide-diol pairs (see below), it is also possible that they arise randomly from experimental or measurement variations. We previously introduced a method to analyze all epoxide-diol ratios simultaneously to infer global sEH activity, named SAMI [23], which improved power in detecting changes in sEH activity in animals and humans when compared to individual ratio estimates. SAMI integrates all changes in epoxide-diol ratios into one index on each condition, reducing the risk of false positives, and the present study shows that SAMI-derived sEH activity or integrated epoxide-diol ratio was not altered by individual oil infusions (Figure 4). We confirmed that sEH activities or protein levels, directly measured in the liver, were not altered by the oil infusions despite variations in plasma FFAs, validating the results of the SAMI approach of integrating all available epoxide-diol ratios for sEH activity.

Other evidence that FFA availability does not significantly interfere with the use of epoxide-diol ratios as indices of sEH activity come from our previous study [24]. This study showed that, after feeding, plasma FFA epoxides increased (LA, Figure 7), were unaltered (ARA), or decreased (ALA, EPA, DHA), probably reflecting changes in circulating (or intracellular) levels of their precursor FFAs, but epoxide-diol ratios showed similar changes, regardless of their origins (i.e., precursor FFAs). Thus, precursor FFAs might exert mass-action effects to alter epoxide and diol levels, but these effects similarly impacted epoxide and diol levels, not altering epoxide-diol ratios, supporting the concept that epoxide-diol ratios may reflect sEH activity independent of circulating or intracellular levels of their precursor FFAs (or any factors altering FFA epoxides, such as CYP activity).

Sometimes having ratios of metabolites can provide insight overlooked by the absolute numbers. In this case, these ratios have provided great insight into lipid biology. However, there are many factors that go into the epoxide-diol ratios making this ratio very valuable but also difficult to interpret. It should be emphasized that the fatty acid epoxides and diols have vastly different polarities and will be in very different pools in an animal. The ratio implies that the metabolism of these epoxides and diols is simple and dominated by a single biosynthetic and hydrolytic pathway. In fact, the ratio encompasses multiple enzymes involved in biosynthesis, the release of not only precursor lipids but also preformed epoxides from phospholipid pools, competing pathways of degradation, and re-sequestration, to name a few [7,24]. Also, the ratio number is very informative, but the variation can be very high since it is dependent on the confidence intervals of both the numerator and denominator; the variation in the ratio is going to be far greater and the confidence far less than the absolute variation in the denominator and numerator individually. Thus, the ratio provides great insight but at the cost of obscuring the absolute values of epoxides and diols and reducing statistical confidence in your ratio data. Despite these concerns, epoxide-diol ratios often show very robust changes, and when multiple diol-epoxide pairs show similar changes in their ratios, such changes can be attributed to changes in sEH activity, which is the common denominator of factors affecting these ratios [24].

The present study demonstrates that acute elevation of plasma FFAs exerted substrate effects to increase epoxides and diols but no apparent effects on CYP or sEH activities. Because we only looked at these effects following 3 h of FFA elevation, we cannot exclude the possibility that elevation of plasma FFAs for a longer period or in chronic situations may exert non-mass-action effects on CYP and/or sEH activities by altering their protein expression or other factors. In fact, CYP and sEH expression in the liver or other tissues were shown to be altered by high-fat feeding, and these effects required several weeks to develop in rodents [25,26,27,28]. However, when sEH activity is altered with high-fat diets (or chronically elevated FFA levels), this may be reflected as changes in epoxide-diol ratios, which can be attributed, based on the present finding, to changes in sEH activity rather than altered FFA levels (or substrate effects).

Substrate effects of plasma FFA were also observed with hepatic oxylipins produced by LOX and COX. For example, increased plasma LA levels with safflower oil infusion were associated with increased hepatic levels of 9-HODE (Figure 6), produced from LA by 5-LOX. In addition, the elevation of plasma EPA with fish oil infusion increased prostaglandin D_3_ (PGD3), which is produced from EPA by COX, and 5-, 8-, 9-, 13-, and 15-hydroxyeicosapentaenoic acids (HEPEs), which are produced from EPA by 12/15-LOX. The ω-6 FFA LA is known to induce inflammation and insulin resistance, and some of these effects may be mediated by HODEs, such as 9- and 13-HODEs [29,30]. In contrast, increased consumption of the ω-3 FFA EPA improves insulin resistance and metabolic syndrome, and evidence exists that some of these effects are mediated by its metabolites, HEPEs [31,32]. Thus, the effects of dietary ω-3 and ω-6 FFAs (or their ratios) on glucose homeostasis and metabolic syndrome may depend on their substrate effects to produce HODEs vs. HEPEs, rather than their systemic effects to alter other cellular functions (e.g., insulin secretion).

## 4. Materials and Methods

### 4.1. Animals and Catheterization

Male Wistar rats weighing 280–300 g (approximately 9 weeks old) were obtained from Envigo and were housed under controlled temperature (22 ± 2 °C) and lighting (12 h light, 6 a.m.–6 p.m.; 12 h dark, 6 p.m.–6 a.m.) with free access to water and standard rat chow. At least 4 days before the experiment, the animals were placed in individual cages with tail restraints, as previously described [21,22], which were required to protect tail blood-vessel catheters during the experiments. The animals were free to move about and were allowed unrestricted access to food and water. A tail-vein catheter for an infusion and a tail-artery catheter for blood sampling was placed in the evening prior to and in the morning (~7 a.m.) of the experiment, respectively. All procedures involving animals were approved by the Institutional Animal Care and Use Committee at the University of Southern California.

### 4.2. Animal Experiments

Starting at ~1 p.m., after a brief 6 h fast, animals received a constant intravenous infusion of olive (Sigma-Aldrich, St. Louis, MO, USA; O-1514), safflower seed (Sigma-Aldrich, S-8281), or fish (Sigma-Aldrich, F-8020) oil for 2 or 3 h. Control rats received an infusion of vehicle solution (see below). The purpose of these oil infusions was to selectively raise circulating FFAs. Our previous study [21,22] showed that olive oil, which is mainly (~70%) composed of OA (C18:1), selectively raised plasma OA level, and safflower seed oil, mainly (~72%) composed of LA (C18:2), selectively raised plasma LA level. In addition, fish oil was infused to raise circulating levels of ω-3 FFAs, such as DHA (C22:6) and EPA (C20:5). The oils were infused as an emulsion (20% *w*/*v*.(see below), 0.5 mL/h), together with heparin (40 U/h with a bolus injection of 10 U) for stimulation of lipolysis of the infused oils. Oil emulsion was prepared as described by Stein et al. [33]. Briefly, glycerol (2.5% *w*/*v*, 24 mL), phosphatidylcholine (0.36 g; Sigma Chemicals, St. Louis, MO, USA), penicillin–streptomycin solution (0.3 mL; Sigma Chemicals), and either olive, safflower seed, or fish oil (6 g) were placed in a 50 mL tube. After heating at 80 °C for 10 min, the mixture was sonicated three times for 1 min (with 1 min pauses) at a power setting of seven on a Branson Sonifier 450 (Branson Ultrasonics, Danbury, CT, USA). The emulsions were kept at 4 °C and used within 3 or 4 days. The vehicle solution included all of the components except for oil.

Blood samples for the determination of plasma FFAs were collected using the tail-artery catheter before and/or after the lipid infusions. Liver samples were collected after perfusion with saline to avoid the contamination of liver samples with blood. Blood samples were rapidly spun, and plasmas were isolated; plasmas for oxylipin analysis were mixed with triphenylphosphine (TPP; 4 µg/mL), butylated hydroxytoluene (BHT; 4 µg/mL), and EDTA (20 µg/mL) [34]. TPP was used to reduce peroxides to their monohydroxy equivalent, and BHT was used to quench radical-catalyzed reactions. Plasma and liver samples were frozen immediately in liquid N_2_ and stored at −80 °C until analysis.

### 4.3. Analysis

Plasma and liver samples were analyzed for oxylipins using procedures previously reported [34,35]. Individual plasma FFAs were determined using GC-TOF MS by the Metabolomics Service Core of the West Coast Metabolomics Center at UC Davis. Plasma total FFA was measured using an acyl-CoA oxidase-based colorimetric kit from Wako Chemicals Inc. (Richmond, VA, USA). sEH activity was determined in liver extracts by using a radiometric assay with [^3^H]-*t*DPPO as substrate, following described procedures [36]. Hepatic sEH protein level was determined by sandwich ELISA using polyclonal antibody-nanobody as a pair and SA-PolyHRP as the tracer, as described by Li et al. [37].

### 4.4. Statistical Analysis

All data are expressed as means ± S.E.M. The significance of differences in the mean values was assessed by Student’s *t*-tests or one-way ANOVA followed by ad hoc analysis using the Bonferroni method for multiple comparisons. A *p*-value less than 0.05 was considered to be statistically significant.

## 5. Conclusions

The present study showed that raising plasma FFAs exerted substrate effects to increase hepatic and plasma epoxides and diol levels. These increases in epoxides and diols occurred to similar extents, resulting in no significant changes in epoxide-diol ratios. These data suggest that epoxide-diol ratios, often used as indices of sEH activity, are not affected by substrate availability or altered plasma FFA levels and that epoxide-diol ratios may be compared to reflect sEH activity between conditions with different circulating FFA levels.

## Figures and Tables

**Figure 1 ijms-24-10760-f001:**
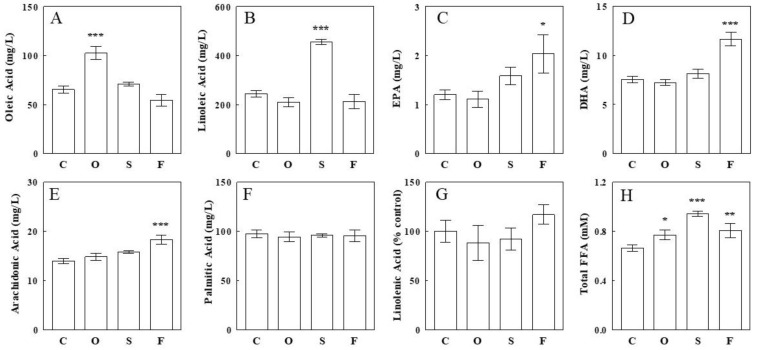
Effects of an intravenous infusion of olive (O), safflower seed (S), or fish (O) oil on individual (**A**–**G**) and total (**H**) plasma FFA levels. Oils were infused as emulsions (see the Section 4) at an infusion rate of 100 mg/h for 2 h. Data are means ± SE (*n* = 6 for each group). *, *p* < 0.05; **, *p* < 0.01; ***, *p* < 0.001 vs. control (**C**).

**Figure 2 ijms-24-10760-f002:**
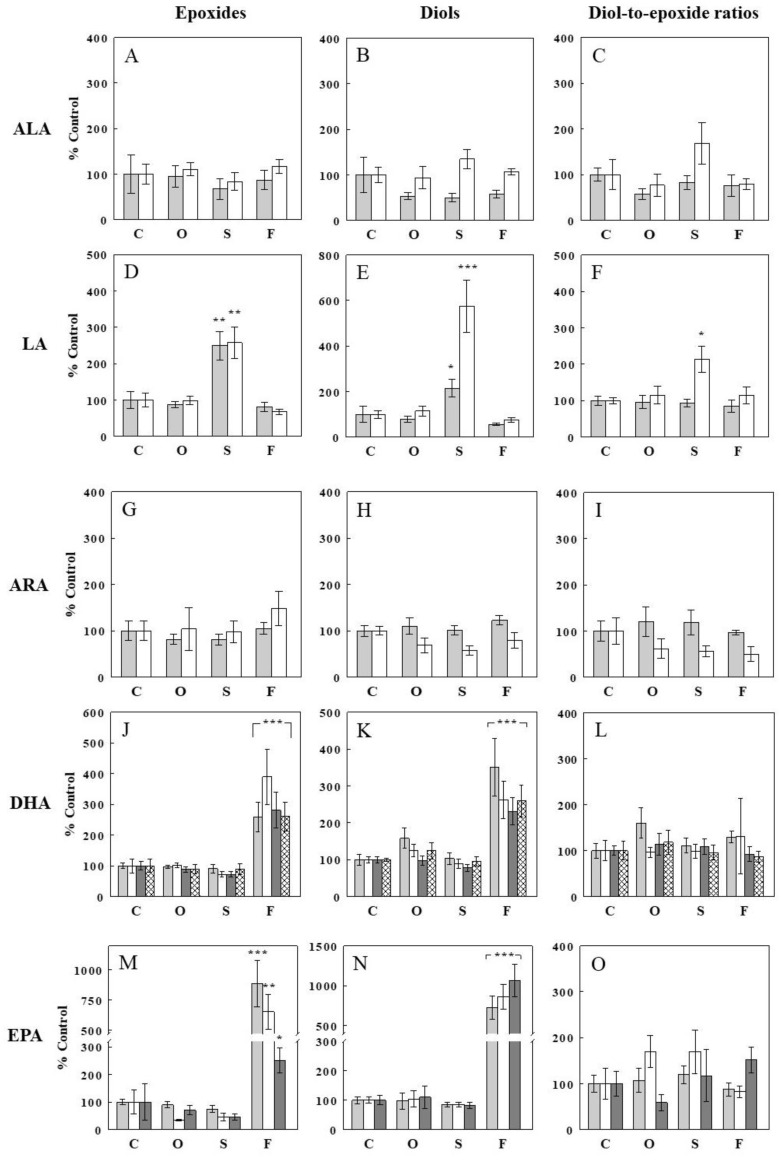
Effects of an intravenous infusion of olive (O), safflower seed (S), or fish (F) oil on hepatic levels of epoxides (**A**,**D**,**G**,**J**,**M**), diols (**B**,**E**,**H**,**K**,**N**), and diol-to-epoxide ratios (**C**,**F**,**I**,**L**,**O**) derived from precursor FFAs, i.e., α-linolenic acid (ALA; **A**–**C**), linoleic acid (LA; **D**–**F**), arachidonic acid (ARA; **G**–**I**), docosahexaenoic acid (DHA; **J**–**L**), eicosapentaenoic acid (EPA; **M**–**O**). Data are means ± SE (*n* = 5 or 6 for each group), expressed as % control (**C**) (see Appendix A for absolute levels of epoxides, diols, and diol-to-epoxide ratios). *, *p* < 0.05; **, *p* < 0.01; ***, *p* < 0.001 vs. control. Epoxides derived from ALA are 15(16)-EpODE and 9(10)-EpODE (from left); from LA, 12(13)-EpOME and 9(10)-EpOME (from left); from ARA, 11(12)-EpETrE and 5(6)-EpETrE (from left); from DHA, 10(11)-EpDPE, 13(14)-EpDPE, 16(17)-EpDPE, and 19(20)-EpDPE (from left); from EPA, 14(15) EpETE, 11(12) EpETE, and 8(9) EpETE (from left). The corresponding diols are 15,16-DiHODE and 9,10-DiHODE (ALA); 12,13-DiHOME and 9,10-DiHOME (LA); 11,12-DiHETrE and 5,6-DiHETrE (ARA); 10,11-DiHDPE, 13,14-DiHDPE, 16,17-DiHDPE, 19,20-DiHDPE (DHA); 14,15-DiHETE, 11,12-DiHETE, and 8,9-DiHETE (EPA).

**Figure 3 ijms-24-10760-f003:**
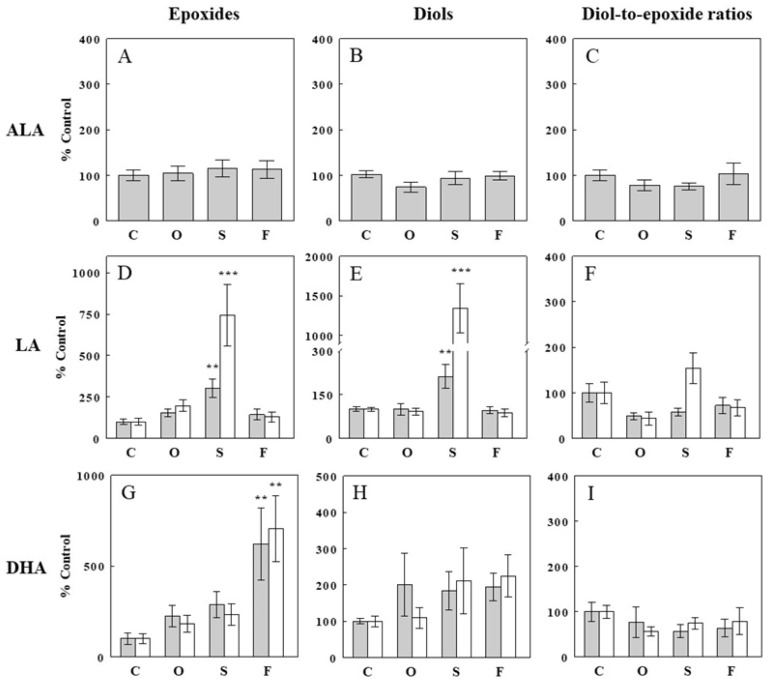
Effects of an intravenous infusion of olive (O), safflower seed (S), or fish (F) oil on plasma levels of epoxides (**A**,**D**,**G**), diols (**B**,**E**,**H**), and diol-to-epoxide ratios (**C**,**F**,**I**) derived from precursor FFAs, i.e., α-linolenic acid (ALA; **A**–**C**), linoleic acid (LA; **D**–**F**), and docosahexaenoic acid (DHA; **G**–**I**). Data are means ± SE (*n* = 10–12 for each group), expressed as % control (**C**) (see Appendix A for absolute levels of epoxides, diols, and diol-to-epoxide ratios). **, *p* < 0.01; ***, *p* < 0.001 vs. control. Epoxides shown are 15(16)-EpODE (from ALA); 12(13)-EpOME and 9(10)-EpOME (from LA; from left); and 16(17)-EpDPE and 19(20)-EpDPE (from DHA; from left). The corresponding diols are 15,16-DiHODE (ALA); 12,13-DiHOME and 9,10-DiHOME (LA); 16,17-DiHDPE, 19,20-DiHDPE (DHA).

**Figure 4 ijms-24-10760-f004:**
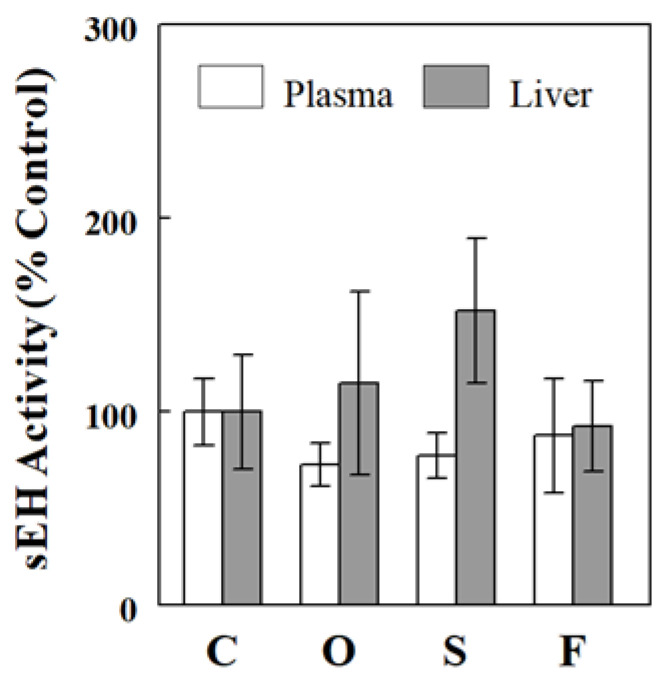
sEH activity estimated by SAMI based on plasma or hepatic epoxide-diol ratios. Data are means ± SE (*n* = 5 or 6 for the liver and 12 for plasma). C, control; O, olive oil; S, safflower seed oil; F, fish oil.

**Figure 5 ijms-24-10760-f005:**
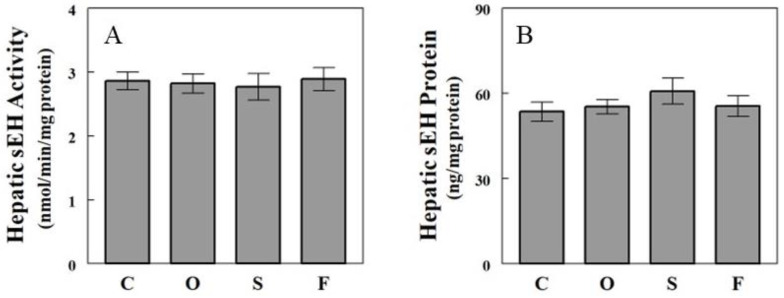
Effects of oil infusions on sEH activity (**A**) and protein level (**B**) measured in the liver, as described in the Materials and Methods section. Data are means ± SE (*n* = 6). C, control; O, olive oil; S, safflower seed oil; F, fish oil.

**Figure 6 ijms-24-10760-f006:**
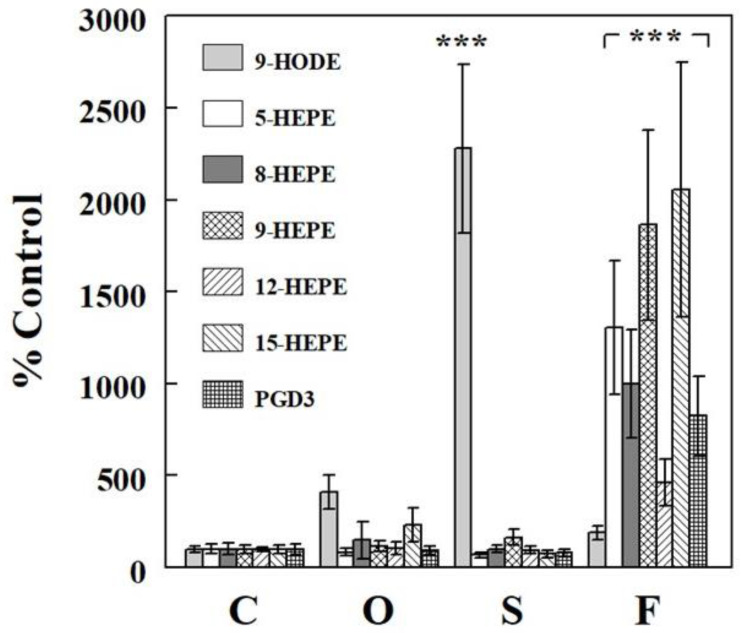
Effects of oil infusions on hepatic levels of oxylipins produced by LOX and COX, derived from LA (9-HODE) and EPA (HEPEs and PGD3). Data are means ± SE (*n* = 3–6 for each group), expressed as % control (see Appendix A for absolute levels of oxylipins). ***, *p* < 0.001 vs. control. C, control; O, olive oil; S, safflower seed oil; F, fish oil.

**Figure 7 ijms-24-10760-f007:**
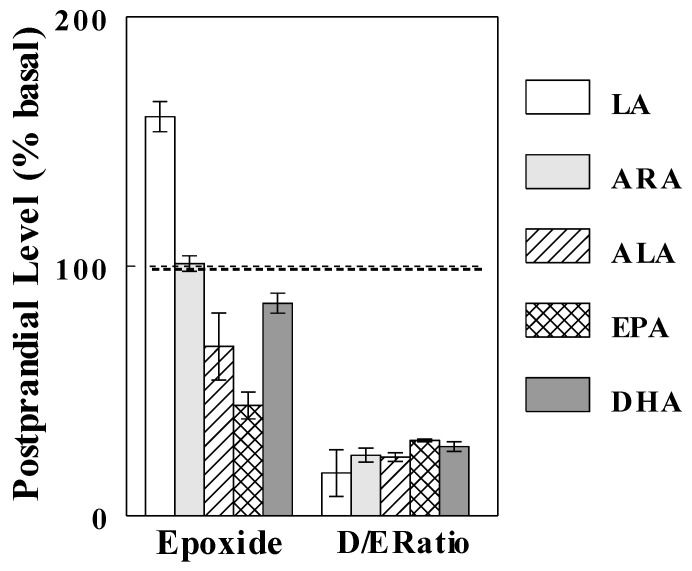
Postprandial levels of FFA epoxides and diol-to-epoxide (D/E) ratios derived from different precursor FFAs (data from Ref. [24]). Data are means ± SEM (*n* = 6).

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
