# Peer review of "Effects of Individual Circulating FFAs on Plasma and Hepatic FFA Epoxides, Diols, and Epoxide-Diol Ratios as Indices of Soluble Epoxide Hydrolase Activity"

_ijms, 2023, doi:10.3390/ijms241310760_

Round 1

Reviewer 1 Report

In their current study, the authors examined the effects of acute elevation of individual plasma FFAs on oxylipins, particularly, epoxides and diols and their ratios. As an in vivo model, Wistar rats infused with emulsions of olive and thistle fish oil via the tail vein for 2 or 3 h were used. Individual plasma FFAs were determined using GC-TOF MS and the epoxides and diols formed also using MS techniques. The oil-infusions raised plasma increased FFA concentrations (OA (+57%), LA (+87%), EPA (+70%) DHA (+54%)).

The elevated plasma FFAs excerted substrate effects to increase hepatic and plasma epoxide and diol levels but no significant changes in epoxide-diol ratios. The authors concluded from their findings that the soluble epoxide hydrolases (sEH) activity are not affected by substrate availability or altered plasma FFA levels thus the epoxide-diol ratios may to reflect sEH activity.

 The question addresseby the authors, whether the sEH activity is dependent on the substrate concentration, is relevant for the scientific community and of fundamental importance for the implementation of experiments with sEH and their interpretation. The straight forward study design leads to significant results and clear conclusions. The authors have already refuted my initial marginal criticism of the study in the discussion, in which they once again explicitly pointed out that in the present study only the possible short-term effects of increased FFA concentrations on sEH activity were examined (2 or 3 h ) and that it is quite possible that for example increased plasma FFA concentration for several days can influence the sEH activity.

There is a typo in line 336: Instead of "radio" it must be "ratio".

Author Response

We thank the referee for careful evaluation of the manuscript, appreciating the significance of the study, and providing positive comments.

We also thank the referee for finding the typo, which is now fixed. 

Reviewer 2 Report

This manuscript is titled “Effects of Individual Circulating FFAs on Plasma and Hepatic FFA Epoxides, Diols, and Epoxide-diol Ratios as Indices of Soluble Epoxide Hydrolase Activity”. However, it does not read well. The format and structure of the manuscript are problematic. I highly recommend the authors address these issues first before submitting the manuscript again. Here are my comments:

1.       The resolution of most of the figures is low and needs to be improved.

2.       In Figure 2, the figure legend is missing, the authors should specify what the gray and empty represents, and each graph needs to be marked with letters. They need to be placed beside the figure.

3.       There is no figure 3 but has figures 4, 5, 6, and 7.

4.       The authors need to point out which figure supports the result 3.3.

5.       Figure 7 was not described in the result section.

Author Response

We thank the referee for careful evaluation of the manuscript. Figure 3 and its legend were missing during the conversion by the editorial office of our submitted manuscript to the “adjusted” manuscript that was provided to the referees. Figure 3 and its legend are now included in the revised manuscript. In addition, we tried to improve the readability of the manuscript, especially the Results section, by labeling subfigures of Figures 2 and 3 and providing references to these subfigures in the main text.

  1. We now provide all figures in high resolution.
  2. Figure 2 legend is now included, and all subfigures are marked with letters.
  3. Figure 3 was missing and is now included. Figures 4-7 remain the same.
  4. Figure 3 presents data described in the result 3.3 (now 2.3) section. We now provide references in the text to subfigures of Figure 3.
  5. Figure 7 shows data from a previously published study, which were described and discussed in the Discussion section.

Round 2

Reviewer 2 Report

The authors have addressed my questions.